# First Case Report of Developmental Bilateral Cataract with a Novel Mutation in the *ZEB2* Gene Observed in Mowat-Wilson Syndrome

**DOI:** 10.3390/medicina59010101

**Published:** 2023-01-02

**Authors:** Agnieszka Tronina, Marta Świerczyńska, Erita Filipek

**Affiliations:** 1Department of Paediatric Ophthalmology, Faculty of Medical Sciences, Medical University of Silesia, Kornel Gibiński University Clinical Center, 40-055 Katowice, Poland; 2Department of Ophthalmology, Faculty of Medical Sciences, Medical University of Silesia, Kornel Gibiński University Clinical Center, 40-055 Katowice, Poland

**Keywords:** Mowat Wilson syndrome, MWS, developmental cataract, Smad interaction protein 1, SIP 1, *ZEB2* gene

## Abstract

*Background*: Mowat-Wilson syndrome (MWS) is extremely rare multisystemic autosomal dominant disorder caused by mutations in the Zinc Finger E-Box Binding Homeobox 2 (*ZEB2*) gene. Ocular pathologies are one of the symptoms that appear in the clinical picture of MWS individuals, but not many have been described so far. Pathologies such as optic nerve or retinal epithelium atrophy, iris or optic disc coloboma as well as congenital cataracts have been most frequently described until now. Therefore, we would like to report the first case of bilateral developmental cataract in a 9-year-old girl with MWS who underwent successful cataract surgery with intraocular lens implantation. *Case Presentation*: A 9-year-old girl, diagnosed with p.Gln694Ter mutation in *ZEB2* gene and suspicion of MWS was referred to the Children’s Outpatient Ophthalmology Clinic for ophthalmological evaluation. Her previous assessments revealed abnormalities of the optic nerve discs. The patient was diagnosed with atrophy of the optic nerves, convergent strabismus, and with-the-rule astigmatism. One year later, during the follow-up visit, the patient was presented with decreased visual acuity (VA), developmental total cataract in the right eye and a developmental partial cataract in the left eye. This resulted in decreased VA confirmed by deteriorated responses in visual evoked potential (VEP) test. The girl underwent a two-stage procedure of cataract removal, first of one eye and then of the other eye with artificial lens implants. In the 2 years following the operation, no complications were observed and, most remarkably, VA improved significantly. *Conclusions*: The *ZEB2* gene is primarily responsible for encoding the Smad interaction protein 1 (SIP1), which is involved in the proper development of various eye components. When mutated, it results in multilevel abnormalities, also in the proper lens formation, that prevent the child from normal vision development. This typically results in the formation of congenital cataracts in children with MWS syndrome, however, our case shows that it also leads to the formation of developmental cataracts. This is presumably due to the effect of the lack of SIP1 on other genes, altering their downstream expression and is a novel insight into the importance of the SIP1 in the occurrence of ocular pathologies. To the best of our knowledge, this is the first case of bilateral developmental cataract in the context of MWS. Moreover, a novel mutation (p.Gln694Ter) in the *ZEB2* gene was found corresponding to this syndrome entity. This report allows us to gain a more comprehensive insight into the genetic spectrum and the corresponding phenotypic features in MWS syndrome patients.

## 1. Introduction

Mowat-Wilson syndrome (MWS) is a rare, multisystemic inherited disorder characterized by facial dysmorphia, psychomotor retardation, and other congenital malformations of multiple organs. It was first described in 1998 by Mowat et al. and since then, approximately 350 cases have been described worldwide [1]. The variety of clinical manifestations that distinguish this disease entity makes it difficult to make a uniform diagnosis; therefore, many cases are undiagnosed or diagnosed later in the child’s life. The prevalence of MWS is estimated to be 1:50,00–100,000 live-born children and to date, there has been no reported case of affected individuals having offspring. Siblings are rarely reported in the literature, but their very existence may indicate possible germ cell lineage involvement and a mosaic pattern of inheritance between parents and offspring [2,3].

MWS results from heterozygous mutations or deletions in the zinc finger E-box-binding homeobox 2 (*ZEB2*) gene, encoding Smad Interacting Protein 1 (SIP1), which plays a key role in the transforming growth factor-beta (TGF-β) signaling pathway. This cascade is a major intracellular pathway in the human body with distinct effects on cell proliferation and differentiation. The expression of TGF-β family members is elevated during embryonic life, ensuring the promotion of normal organogenesis, including the eye [4,5]. SIP1 expression is higher in cells derived from the neural crest, genital gland, musculoskeletal system, and eyeball. Hence, the localization and eventual damage of these cells directly correspond to the spectrum of malformations presented in MWS [6].

Ocular pathologies are one of the symptoms that appear in the clinical picture of MW individuals, but not many have been described so far. Pathologies such as optic nerve or retinal epithelium atrophy, iris or optic disc coloboma as well as congenital cataracts have been most frequently described until now. Therefore, we would like to report the first case of bilateral developmental cataract in a 9-year-old girl with MWS who underwent successful cataract surgery with intraocular lens (IOL) implantation.

## 2. Case Report

A 9-year-old girl, diagnosed with p.Gln694Ter mutation in *ZEB2* gene and suspicion of MWS was urgently referred to the Children’s Outpatient Ophthalmology Clinic to expand the diagnosis of visual pathway conduction disorders and further evaluation. Her previous ophthalmic assessments revealed pallor of the temporal part of optic nerve discs. The patient was also diagnosed with convergent strabismus and astigmatism.

On admission, the patient’s mother presented her current medical records. The girl was born from her mother’s first pregnancy and first delivery, the birth was by natural force in the 39th week of pregnancy, with a birth weight of 3440 g, body length 52 cm and head circumference 33 cm. The course of pregnancy and delivery complied with the norm. In the postnatal period, a patent ductus arteriosus and aortic valve stenosis were discovered. Furthermore, the occurrence and persistence of constipation led to the initiation of the patient’s multispecialty diagnostics and the diagnosis of Hirschsprung’s disease. Subsequently, due to small increase in head circumference and a noticeable delay in psychomotor development (the child does not establish contact with the environment, does not communicate, nor walk) as well as epileptic seizures the child underwent a number of diagnostic tests including head Magnetic Resonance Imaging (MRI), which revealed callous body dysgenesis, lateral and III ventricular dilatation.

Due to the coexistence of the above clinical manifestations along with facial dysmorphic features characteristic for the Mowat-Wilson syndrome, the child was referred for consultation at the Genetic Clinic. Molecular testing of the coding fragment of the *ZEB2* gene (exon fragment from amino acid position His655 to Ser840) detected a mutation of p.Gla694Ter in one allele. The above mutation involves a C nucleotide substitution at T at position c.2080, resulting in a change in the amino acid Gln to the STOP codon at position 694 of the amino acid chain, which ultimately leads to premature termination of translation. This type of mutation within the *ZEB2* gene has not yet been described in relation to the diagnosis of MWS but in combination with the clinical picture the results spoke for it. In the further later years, epileptic seizures, premature puberty, hypertension, and significant mental and developmental disabilities appeared. The patient’s father (35 years old) is healthy, her mother (30 years old) is treated for hypertension, the girl has no siblings, and the family pedigree is not burdened with genetically determined diseases. Molecular analysis of the *ZEB2* gene performed on the parents did not reveal the presence of pathogenic variants, which means that the mutation in our patient occurred de novo, and the risk of the disease in subsequent children of her biological parents is low (Figure 1).

During the first visit, the patient’s previous ophthalmology medical records were reviewed. The patient did not acquire the ability to speak and, therefore, the visual acuity (VA) could not be verified. Nevertheless, the patient followed objects with both of her eyes and the pupillary direct and indirect response to light was normal. The eyeballs remained in alternating convergent strabismus, Hirschberg test 15–20°, no limitation in eye movements was detected (Figure 2). 

The cycloplegic autorefraction examination revealed in the right eye (RE) −0.5 Dsph −6.0 Dcyl ax 178° and in the left eye (LE) showed −1.0 Dsph −6.5 Dcyl ax 162°. Anterior segment examination of both eyes was characterized by minor posterior synechiae, the optic media were translucent, and funduscopic examination confirmed the previous ophthalmologic findings and showed pallor of the optic nerve discs on the temporal side as well as a subtle inferior elevation of the optic disc in the RE. Ocular ultrasound (USG) showed a slight increase in echogenicity in the area of the optic disc, which was most likely consistent with optic nerve drusen. The patient was scheduled for ophthalmologic follow-up with a prescription of spectacle correction appropriate for her refractive error. 

Less than a year later, the patient had a second visit in our outpatient clinic. According to her mother’s statement, for some time the girl showed worse interest in her surroundings. The patient’s RE did not follow objects (neurologic and developmental causes related to the underlying disease cannot be ruled out), while LE was following objects. The pupil response to indirect and direct light, as well as the swinging light test were normal. Anterior segment eye examination revealed a total cortical cataract in the RE and a partial cortical cataract in the LE. Evaluation of the RE fundus was impossible due to total cataract, fundus of the LE was poorly visible with noticeable optic nerve rim pallor. The USG examination of the RE showed increased echogenicity in the projection of the optic nerve disc and increased echo of the lens, while in the LE it was normal (Figure 3). An FVEP was performed to assess the patient’s visual potential, but due to very difficult cooperation, it was only possible to obtain a record from the RE (Figure 4).

Subsequently, cataract surgery was performed with implantation of a foldable IOL in the RE. In addition, fundus examination of the RE revealed optic nerve pallor with pigment regrouping at the retinal periphery. An identical procedure was then performed in the LE (Figure 5). Residual astigmatism was corrected with spectacles. 

During the follow-up, the patient was able to follow the objects again and the FVEP study results were within normal ranges. The patient is currently undergoing postoperative 3-year follow-up and further ophthalmologic care.

## 3. Discussion

MWS is a disease entity in which there are structural abnormalities of the eye and its various structures including the optic nerve and thus the pathway of conduction of visual impulses. It has been observed that the clinical features of MWS are interdependent on the type of genetic mutation. Patients with gene deletions are phenotypically similar [2]. Some studies have shown that the larger the deletion, the greater the clinical impairment and the greater the similarity to typical MWS features. Unfortunately, no more than 5% of cases are caused by missense mutations of the *ZEB2* gene and thus reveal a mild form of disability [7,8,9]. The morphological features characteristic of MWS patients are shown in table (Table 1). Despite the abnormalities specific to individual systems, the overall impression is important, given that the patient is presenting for the first time. Common emblematic features are observed in 9 out of 10 cases and include a peculiar facial phenotype and posture that become more prominent with the increasing age of the patient. The face tends to be elongated and the nose is extended downward over the philtrum. The upper face is unique with epicanthic folds, orbital hypertelorism, and large, medially arched eyebrows. Characteristics of the midface include a wide nasal bridge and a saddle-shaped nose. Other suggestive features include a triangular, pointed chin, an M-shaped mouth, and unique raised earlobes with a central indentation as shown in Figure 6.

Most patients affected by this condition have short stature. Gait limitations are usually due to central nervous system anomalies and intellectual disability. Some patients remain non-ambulatory throughout life. The sucking habit, that disappears in healthy infants, in children affected by this condition persists throughout life [7,10]. Although a few described cases have developed mildly limited speech abilities, the vast majority have profound communication impairment resulting from sparse language cognition and poor motor control [11]. 

Not many cases of ocular pathologies in MWS were identified. Those that were eventually diagnosed usually involved one eye and no correlation was found with other systemic impairments. According to Ivanovski et al. [12], in a large study carried out in the group of 87 subjects with MWS, about 10% of patients express eye problems. He commonly observed strabismus as a characteristic feature, followed by astigmatism, myopia, nystagmus, and ptosis. There were few patients described with Axenfeld anomaly, iris heterochromia, and retinal coloboma. Apart from those previously mentioned, Kilick et al. [13] presented the case of a patient with iris coloboma, Bourchany et al. [14] describes cataract, atrophy, or absence of optic nerve, while Ariss et al. [15] reports microphthalmia and retinal aplasia (Figure 7). Although cataracts described in children with MWS were all congenital, no developmental cataracts have been reported so far [13,14,15,16]. Several other syndromes are known to manifest several features in common with MWS thereby leading to misdiagnosis [17]. Clinical entities that should be noted include Angelman syndrome, Goldberg-Shprintzen syndrome, Pitt-Hopkins syndrome, Rumbenstein-Taybi syndrome, Smith-Lemli-Optitz syndrome, and CHARGE syndrome. Ocular pathologies are clearly manifested in CHARGE and Goldberg-Shprintzen syndrome. In the former, optic nerve head coloboma, strabismus, and amblyopia are common, while ptosis, hypertelorism, and myopia are described in patients with Goldberg-Shprintzen syndrome [18,19].

SIP1 protein encoded by *ZEB2* gene has direct and indirect influence on retinal cells development. Multiple studies define the development of the retina as a consequence of differentiation of multipotent progenitor cell into the specific progeny cell, while simultaneously, some cells are generated in direct restricted manner [20,21,22]. Cells generated by this process are constituents of different neural retinal layers. Specific retinal cells are differentiated with the cooperation of transcription factors that cooperate together and influence the direction and time of the differentiation [23]. It is the cellular regulator protein by itself, but also interacts either by up or down regulation with other transcription factors. It directly facilitates generation of cells in inner neuronal and controls timing of its differentiation. These mechanisms might explain retinal pathologies arising from *ZEB2* mutation in MWS patients. In the absence or defect of SIP1 protein, the faulty visual stimuli transmission is thought to arise due to loss of bipolar cells function, while rods and cones are not affected [24]. 

Strabismus apparently occurs often in MWS patients. Inappropriate alignment of the eye has a wide range of etiology. In MWS, the presumptive hypothesis is for its epigenetic rather than a genetic origin, related to abnormal development of visual pathway or higher visual structures. 

Concomitant strabismus stands for around 90% of all cases and it is of the central origin with early onset. Improper neurodevelopment evidently leads to abnormal visual perception and strabismus. Such inadequate visual system evolution might be expressed in nystagmus and loss of binocular vision that persists or progress with patient age. Prevalence is due to abnormalities of routing of ganglionic cells and underdeveloped and immature connections between distinct parts of neurons in the visual and proprioceptive pathway; such deficits are described in patients with *ZEB2* mutations [12].

SIP1 expression was extensively implicated in the process of lens development. Early embryogenetic period is presented with SIP1 predominant localization in epithelium and lenticular fiber cells. While lens maturation goes forward, expression of this protein is changed and it is mainly found in the peripheral epithelial and cortical fibers, as well as in previously mentioned retinal layers. Notably, SIP1 protein is crucial in the endothelial to mesenchymal transition process, that results in the separation of the lens from the ectoderm fault in SIP1 is known to disturb cellular processes inside the developing lens before the placode become autonomous vesicle. It involves downregulation of expression of two types of proteins: β- and γ-crystallins and thus alter cellular microsceletone and its desirable elastic and transparent properties [25]. While SIP1 does not appear to regulate lens fiber cell differentiation, lens cell survival, or lens cell proliferation after lens vesicle closure, lack of SIP1 in the lens vesicle results also in faulty migration of fiber cell tips necessary for the proper packing of lens fiber cells and formation of lens sutures. These primary defects in lens development manifest as severe defects in the adult lens fiber cell morphology and organization of the meridional rows, leading to a loss of lens transparency. What corresponds to aforementioned SIP1 expression shift, was verified by Manthey et al. [25] studies on mice with Sip1 knockout. There were two periods in lens morphogenesis implicated by timing of Sip1 loss. While Sip1 is lost on later stages of lens development it presumably alters downstream gene expression as presented in figure (Figure 8) [25,26]. 

Regarding MWS patients that already underwent cataract surgery, expression of SIP1 has additional importance. Two already mentioned processes—epithelial to mesenchymal transition and epithelial lens cells differentiation into fiber cells of lens, if malfunctioning, are known to be fundamental in creation of posterior capsular opacifications. Both are known to be influenced by levels of SIP1, which in case of MWS is variable [27,28]. That might result in frequently observed opacifications of the posterior lens capsule after primary cataract surgery, known as secondary cataract. Thus, cataract extraction alone may not be sufficient in VA restoration and additional procedures, including Neodymium-doped yttrium aluminium garnet (Nd:YAG) laser application, may be required. That also implies the great need of a regular, post-surgical follo- ups, allowing for frequent lens and capsule evaluation. 

Clearly, in the described case, the spectrum of eye abnormalities can appear as developmental but also as acquired conditions. The patient did not present typical, usually congenital rather than developmental triad of eye anomalies. There is no previous report demonstrating developmental cataract in MWS. The accurate understanding of molecular bases that drives cataract development in MWS rise the explanation for this circumstance. Although appropriate means and time were picked, the outcome of cataract surgery cannot be directly compared with otherwise healthy patients. In children with MWS, restoring the best VA by cataract removal can be extremely difficult due to the changes in morphology of the optic nerve and higher central neuronal structures described in MWS, which significantly disrupt the physiology of the visual pathway. 

## 4. Conclusions

Diversity of abnormalities, relevant to ophthalmological evaluation indicates multiple processes that mutation of *ZEB2* gene interferes with. While a variety of potential eye pathologies are discovered and described in the literature, the entire spectrum is probably not yet fully identified and understood. Physicians should take exceptional precautions to regularly evaluate ocular structures and function in MWS patients. Further analysis is needed, comprising the management of specific eye conditions and corresponding outcomes in MWS. That should be of a great interest to rise course of future research, targeting possible molecular therapies of this entity and to create conceivable guideline paths. That will hopefully improve the detection rate of eye problems and allow for an immediate and precise medical approach to patients with MWS.

## Figures and Tables

**Figure 1 medicina-59-00101-f001:**
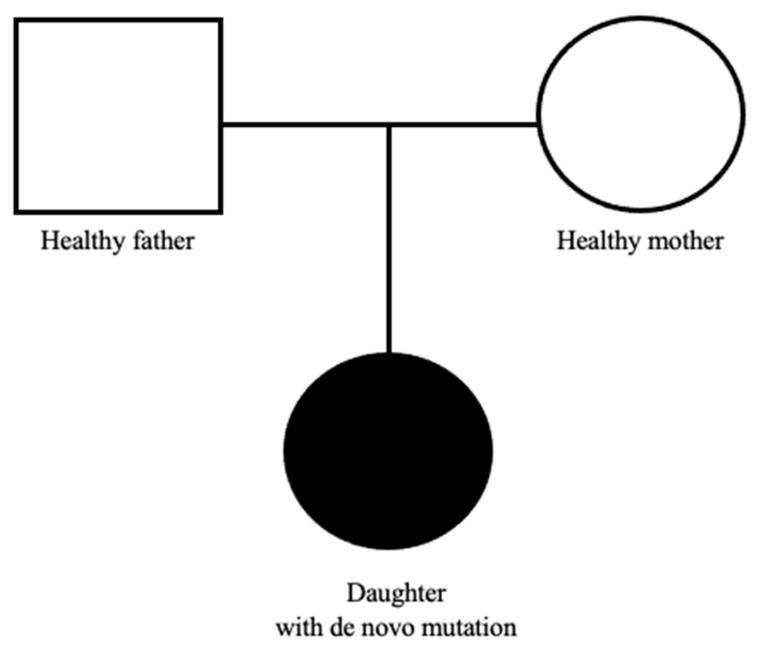
Pedigree chart of our patient.

**Figure 2 medicina-59-00101-f002:**
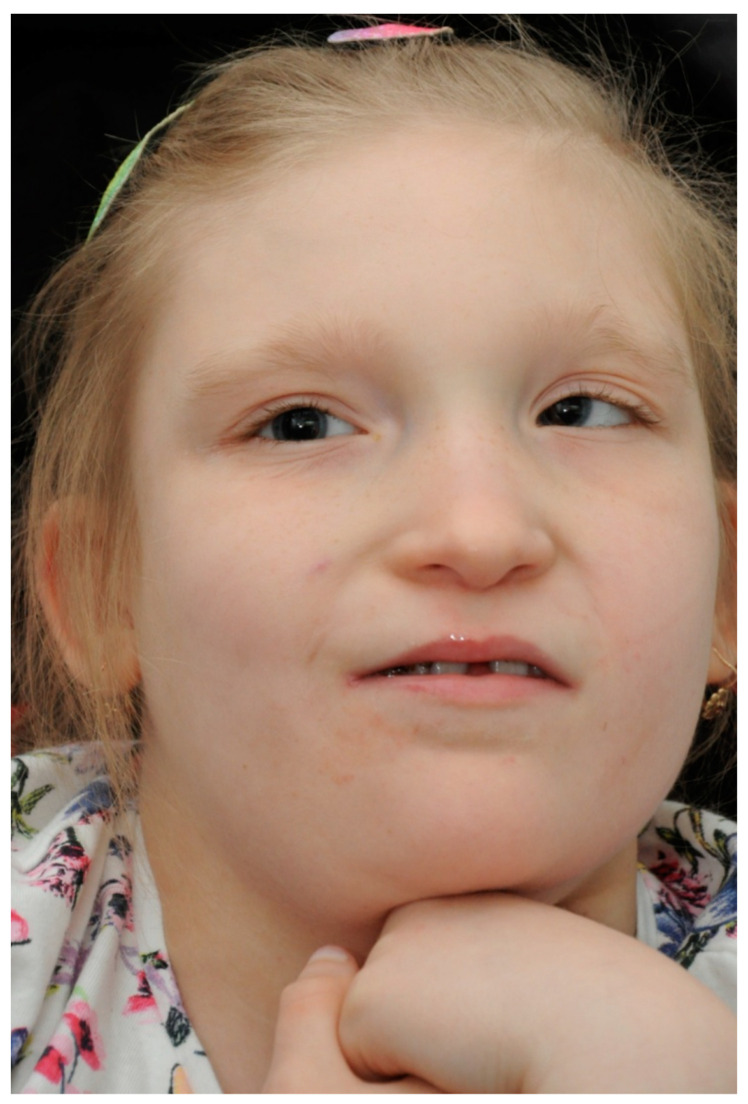
Our patient with convergent strabismus and facial dysmorphia specific to MWS.

**Figure 3 medicina-59-00101-f003:**
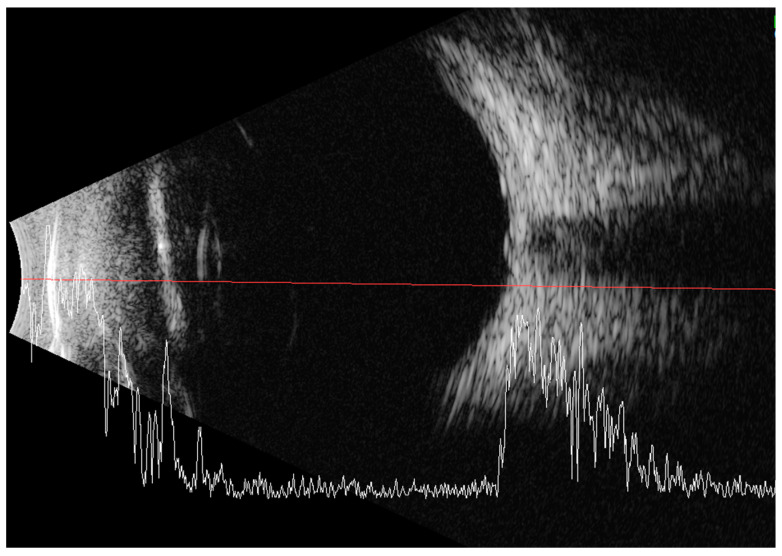
Ultrasound of the RE: increased echogenicity of the lens (cataract) and a slight punctate increase in echogenicity in the projection of the optic disc (which may correspond to external drusen).

**Figure 4 medicina-59-00101-f004:**
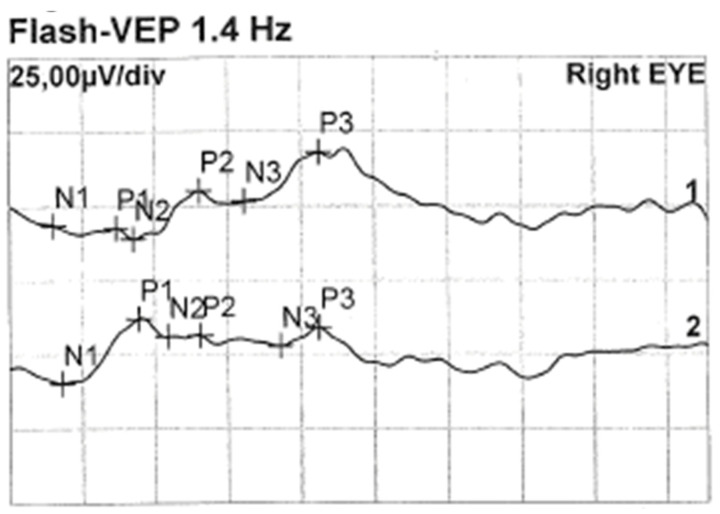
FVEP of the RE: latencies prolonged to 125% of norm. Left hemisphere amplitudes normal, and right hemisphere amplitudes are vestigial.

**Figure 5 medicina-59-00101-f005:**
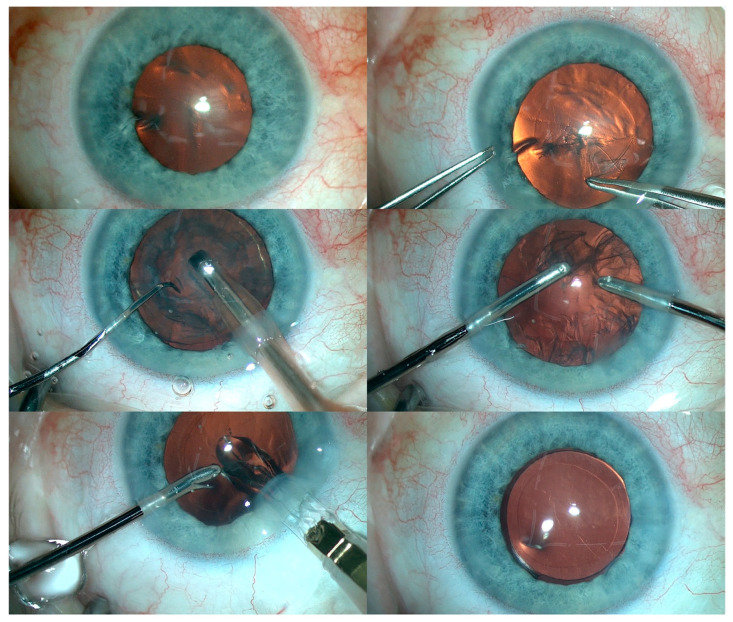
The various steps of cataract removal and intraocular lens implantation in the LE.

**Figure 6 medicina-59-00101-f006:**
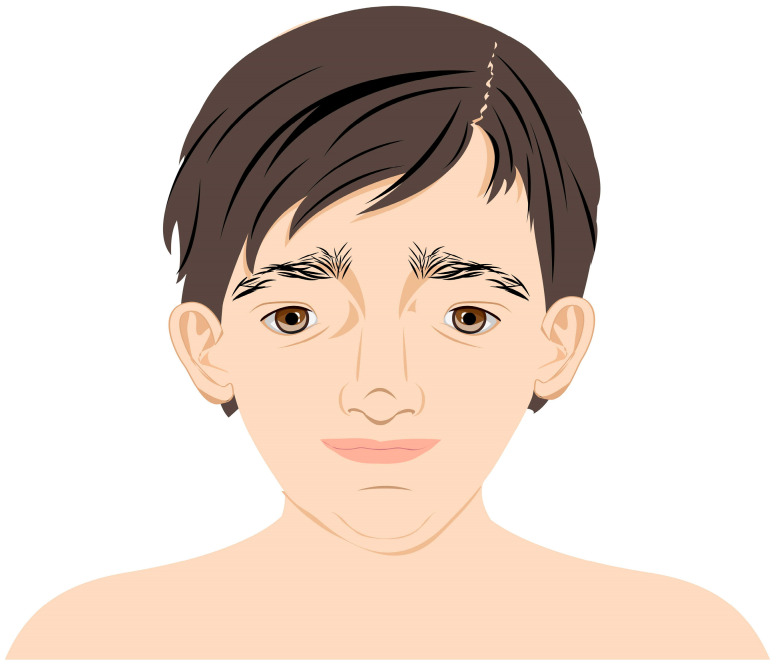
Characteristic features of facial dysmorphia in patients with MWS.

**Figure 7 medicina-59-00101-f007:**
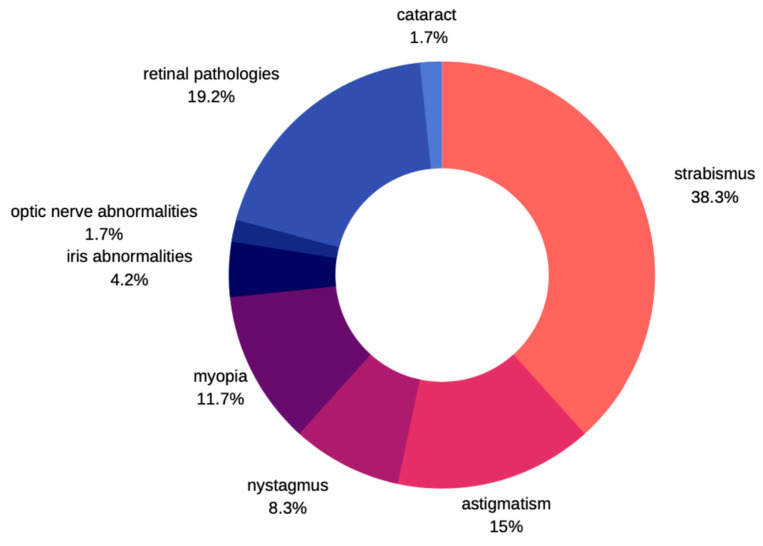
Prevalence of ophthalmic disorders in MWS.

**Figure 8 medicina-59-00101-f008:**
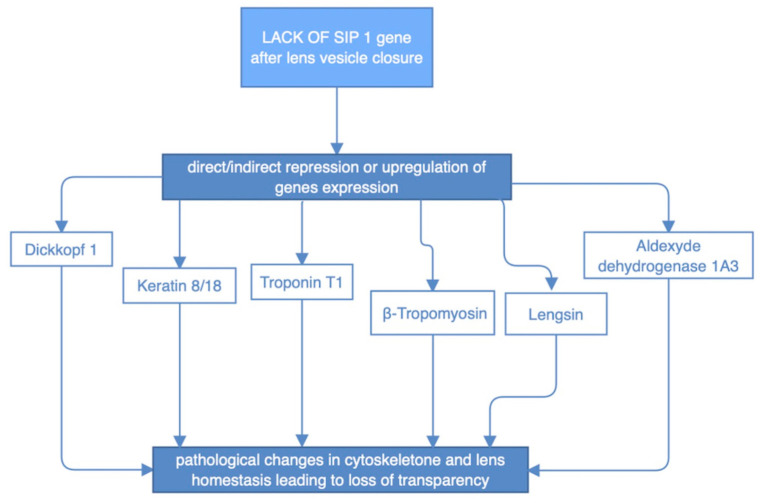
Impact of abnormal SIP1 protein on the expression of certain genes and indirect structural defects of the lens.

**Table 1 medicina-59-00101-t001:** The most frequent abnormalities involving various systems found in MWS.

Organ System	Characteristics
Central Nervous System	Epilepsy, seizures
Gastrointestinal System	Hirschprung disease, constipation, stool incontinence
Cardiovascular System	Structural heart defects more common isolated then multiple: atrioventricular septal defecrs, patent ductus arteriosus, aortic and pulmonary valve stenosis
Genitourinary System	Hydronephrosis, vesicoureteral reflux, urinary incontinence Males: hypospadia, cryptorchidism, bifid scrotum Females: vaginal septum
Musculoskeletal System	Slender fingers, long toes, scoliosis, genu and hallux valgus, pectus excavatum, muscular hypotonia
Ocular System	Strabismus, astigmatism, myopia, nystagmus

## Data Availability

Not applicable.

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
