# Peer review of "First Case Report of Developmental Bilateral Cataract with a Novel Mutation in the ZEB2 Gene Observed in Mowat-Wilson Syndrome"

_medicina, 2023, doi:10.3390/medicina59010101_

Round 1

Reviewer 1 Report

In this paper Tronina et al. described a bilateral cataract case with terminated mutation in ZEB2. Detailed information has been provided on the background of MWS caused by ZEB2. There are still points the authors should improve before be considered for publication.

1. In the Abstract Background: “MWS is extremely rare multisystemic disorder caused by de novo mutations…”. De novo mutation means the mutation is seen only in the patient but not caused by inheritance from the parents. MWS is an autosomal-dominant inherited disease. Here use de novo in general without detailed pedigree information is not appropriate.

2. Similarly, no genetic and phenotypic information of the girl’s parents provided in this article, esp. for the gene ZEB2. There is a lack of family history.

3. In Introduction, “The prevalence of MWS is estimated to be 1:50.00-100.000 live-born…”The number is quite confusing with a dot inside. Should be comma, like 100,000?

4. No detailed clinical examination data was provided for this girl. The authors mentioned the girl had gone through USG, FVEP, ocular imaging with slit lamp etc., so the data and clinical pictures should be provided. especially pictures describing cataract status.

5. Other clinical evidence for the diagnosis of MWS should be provided, such as intellectural assessment, etc.

6. Add reference on Sip1 knockout mice.

Author Response

Thank you for taking the time to review our article and for all your valuable comments. We have highlighted in red color all changes made in the text.

1. In the Abstract Background: "MWS is extremely rare multisystemic disorder caused by de novo mutations...". De novo mutation means the mutation is seen only in the patient but not caused by inheritance from the parents. MWS is an autosomal-dominant inherited disease. Here use de novo in general without detailed pedigree information is not appropriate.

Thank you for pointing out this unintentional shortcut. 

Mowat-Wilson Syndrome (MWS) is an autosomal dominant disorder caused by a pathogenic variant in ZEB2. However, almost all individuals reported to date have been simplex cases (i.e., a single occurrence in a family) resulting from a de novo genetic alteration, Rarely, recurrence in a family has been reported when a parent has a low level of somatic or presumed germline mosaicism for a MWS-causing pathogenic variant. Individuals with MWS are not known to reproduce [1]. 

Adam MP, Conta J, Bean LJH. Mowat-Wilson Syndrome. 2007 Mar 28 [Updated 2019 Jul 25]. In: Adam MP, Everman DB, Mirzaa GM, et al., editors. GeneReviews® [Internet]. Seattle (WA): University of Washington, Seattle; 1993-2022.

We have revised "Abstract Background" and "2. Case report" to clarify the genetic aspect.

2. Similarly, no genetic and phenotypic information of the girl's parents provided in this article, esp. for the gene ZEB2. There is a lack of family history.

We agree with the above suggestion and thank you for it. 

We have supplemented the content of "2. Case report" with additional genetic information.

3. In Introduction, "The prevalence of MWS is estimated to be 1:50,00-100,000 live-born... "The number is quite confusing with a dot inside. Should be comma, like 100,000?

Of course, we agree with the above suggestion. We have made a change to "Intrduction".

4. No detailed clinical examination data was provided for this girl. The authors mentioned the girl had gone through ultrasound, FVEP, ocular imaging with slit lamp etc., so the data and clinical pictures should be provided. especially pictures describing cataract status.

We supplemented the "Case report" with the examinations in our possession, i.e. the ocular usg, the result of the Flash VEP and the photographic documentation of the operation, which also show the appearance of the anterior segment of the eyeball with the cataract and after its removal.

However, it should be borne in mind that the described patient presented a significant delay in psychomotor development and lack of logical contact. It was impossible to establish cooperation with the girl and, consequently, to perform anterior segment photography under a slit lamp without general anesthesia. Therefore, we supplemented the case report only with photographs taken intraoperatively, which, however, show cataract status.

5. Other clinical evidence for the diagnosis of MWS should be provided, such as intellectural assessment, etc.

Thank you for your suggestion. We have supplemented the "Case report" text with additional data according to medcial record.

6. Add reference on Sip1 knockout mice.

The mentioned literature has been supplemented in "Discussion" and "References".

Reviewer 2 Report

I'm really confused, is it case report or review of MWS. I can't imagine that a case report about the eye manifestations without any picture for the eye either preoperatively or postoperatively. As the authors mentioned. they examined the patient before the development of the bilateral cataract and follow up the patient until cataract development within less than a year, did bilateral operation and follow up patient for three years postoperatively without any picture.

In addition, a lot of missing data in eye diagnosis like type of convergent squint, unilateral or alternating, swinging light reflex, etc.

 Which type of VER can be affected with cataract? Flash or pattern?

Can you expect within less than a year total opacification of cataract due to developmental problem? 

Authors must rewrite the manuscript again giving us the full and scientific data of the eye problems with figures.

Author Response

Thank you for taking the time to review our article and for all your valuable comments. We have highlighted the changes made to the manuscript test in red color. 

1. I can't imagine that a case report about the eye manifestations without any picture for the eye either preoperatively or postoperatively. As the authors mentioned. they examined the patient before the development of the bilateral cataract and followed up the patient until cataract development within less than a year, did bilateral operation and followed up patient for three years postoperatively without any picture.

Of course, we agree with the above opinion. We supplemented the "Case report" with the photographic documentation we have of the operation, which also shows the appearance of the anterior segment of the eyeball with and after cataract removal.

However, it should be borne in mind that the described patient presented a significant delay in psychomotor development and lack of logical contact. Establishing cooperation with the girl and, consequently, performing anterior segment photography under a slit lamp without general anesthesia was impossible. Therefore, we supplemented the case report with photographs taken intraoperatively, which show cataract status.

2. In addition, a lot of missing data in eye diagnosis like type of convergent squint, unilateral or alternating, swinging light reflex, etc.

Thank you for this suggestion. We have supplemented the "Case report" text with additional data, as well as the results of the ocular usg and Flash VEP.

3. Which type of VER can be affected with cataract? Flash or pattern?

Cataracts or corneal haze can cause a prolonged P100 latency in Pattern VEP (PVEP). In contrast, the impaired Flash VEP (FVEP) result is more individually variable than the PVEP and usually shapes similarly in both eyes of a given patient. The FVEP can be useful in patients who are uncooperative and in cases where the lack of optic center translucency prevents the use of pattern stimuli [1, 2].

However, it should be borne in mind that PVEP can be performed only when the patient cooperates well and presents with visual acuity > 0.1 according to Snellen [2]. 

The patient we described showed significant psychomotor developmental delay, and due to the lack of speech and logical contact, visual acuity was difficult to establish. For this reason, we attempted an FVEP in the patient to assess her visual potential.  However, the girl's lack of cooperation only allowed us to obtain a recording from the right eye. 

1 Pojda-Wilczek D. Electrophysiologic differential diagnosis of visual disorders. Ophthalmology after the Diploma. 2017, 2.

2. Goslawski W., Lubinski W. Electrophysiological examination in ophthalmic clinical practice. Part I. Diagnostics of the optic nerve and visual pathway. Ophthalmology. 2016, 2, 1-16.

4.Can you expect within less than a year total opacification of cataract due to developmental problem? 

The case we describe of bilateral developmental cataract formation within a year involves a patient diagnosed with mutation in ZEB2 gene. The ZEB2 gene is primarily responsible for encoding the Smad Interaction Protein 1 (SIP1). In turn, SIP1 expression was extensively implicated in the process of lens development. The early embryogenetic period is presented with SIP1 predominant localization in epithelium and lenticular fiber cells. While lens maturation goes forward, expression of this protein is changed and it is mainly found in the peripheral epithelium and cortical fibers (the role played by SIP1 is described by us in "Discussion"). Thus, it can be assumed that the above mutation, by disrupting the normal function of SIP, affects the development of cataracts.

In addition, a p.Gln694Ter mutation in ZEB2 gene was detected in our patient, which involves a C nucleotide substitution at T at position c.2080, resulting in a change of the Gln amino acid to the STOP codon at position 694 of the amino acid chain, and this leads to premature translation termination, i.e. shortening of the protein product. 

This mutation has not been described so far, and its impact on cataract development has not been studied. However, given the above data - there is a possibility that the p.Gln694Ter mutation in ZEB2 gene is associated with baking the process of own lens opacity. 

In the literature one can find, among others, a case of progressive bilateral nuclear cataracts associated with cerebellar-facialdental syndrome in a 16-year-old patient with 1p36 deletion syndrome, who developed bilateral developmental cataracts within 2 months causing a decrease in visual acuity from 20/20 in each eye to 20/200 in the right eye and 20/25 in the left eye. The authors speculate that such rapid cataract formation may be related to a genetic disorder, which, however, has not yet been sufficiently studied [3]. 

3. Danese C, Pignatto S, Lanzetta P. Noncongenital juvenile-onset bilateral lamellar cataract in 1p36 deletion syndrome. J AAPOS. 2021, 25, 368-370. doi: 10.1016/j.jaapos.2021.07.003

Round 2

Reviewer 1 Report

Most of the questions were well addressed. Minor improvements are suggested:

1. Draw a family tree in Figure 1. Though both mother and father are non-MWS and have no ZEB2 mutation, as described in lines 130-131. A pedigree diagram is easier for readers to understand. 

2. Figure 4. Better change the surgical video to the slit-lamp photo before and after surgery. Easier to interpret. 

Author Response

We sincerely thank you for your review and comments sent.

1 Draw the family tree in Figure 1. Although both mother and father are not MWS patients and do not have the ZEB2 mutation, as described in lines 130-131. The pedigree diagram is easier for readers to understand. 

Thank you for the above suggestion. We have added the pedigree diagram to the "Case description".

2 Figure 4: It is better to replace the surgical film with a slit lamp photo before and after surgery. Easier to interpret. 

Of course, we agree that photographic documentation taken at the slit lamp would be most appropriate. Unfortunately, as we mentioned previously, the described patient presented significant psychomotor retardation and lack of logical contact. It was impossible to establish cooperation with the girl and, consequently, to take an anterior slit-lamp photograph of the eye without general anesthesia. Therefore, we supplemented the case description only with photos taken intraoperatively.

Reviewer 2 Report

Thank you for your adding some missing data, but some signs need explanation.

1- Authors mentioned that Rt optic disc is pale denoting Rt optic nerve atrophy although swinging light reflex was normal.

2- Authors added picture of steps of cataract removal but density of cataract is contradicting with total lens opacity they mentioned. Also, why you are using phacoemulsification in a soft cataract?

3- Authors mentioned that Rt fundus examination revealed pallor of optic disc without mentioning any other abnormalities like drusen. So, why in ultrasound there was increased echogenicity at optic disc?

Author Response

Thank you very much for your review and substantive comments. We have highlighted changes made to the text of the manuscript in red.

1- Authors mentioned that Rt optic disc is pale denoting Rt optic nerve atrophy although swinging light reflex was normal.

Thank you for this comment. We assume that the swinging light reflex did not show a relative afferent pupillary defect (RAPD), since the temporal optic disc atrophy of the n.II nerve was symmetrical in the study girl. In addition, although it was possible to obtain an FVEP result from the right eye only, it can be assumed that the changes involving the pattern pathway (resulting from the underlying disease and its complications) in the patient are bilateral. We have made changes to the "Case report" to clarify the above issue. 

2- Authors added picture of steps of cataract removal but density of cataract is contradicting with total lens opacity they mentioned. Also, why are you using phacoemulsification in a soft cataract?

Thank you very much for the above comment and for pointing out this inaccuracy.Of course, the posted photographs of the stages of partial cortical cataract removal refer to the left eye (we have edited the caption of Figure 5. in "Case report"), unfortunately we do not have photographs of the cataract removal surgery of the right eye.

In the described case, during the cataract removal surgery, hard and high density cataract was found, which is occasionally diagnosed in cases of cataract patients resulting from genetic mutations. For this reason, it was decided to perform phacoemulsification with no ultrasonics, followed by phacoaspiration of the cataract masses. 

3- Authors mentioned that Rt fundus examination revealed pallor of optic disc without mentioning any other abnormalities like drusen. So, why in ultrasound there was increased echogenicity at optic disc?

Thank you very much for the above comment and for pointing out this inaccuracy.

In the fundus examination of the girl we described, we found morphological features of the optic disc of n. II that could correspond to external drusen, which was confirmed by ultrasound. We agree that this was not described clearly enough in the case report. We have made changes to the "Case report" description.
